# Mesenchymal Stem Cell Exosomes Derived from Feline Adipose Tissue Enhance the Effects of Anti-Inflammation Compared to Fibroblasts-Derived Exosomes

**DOI:** 10.3390/vetsci8090182

**Published:** 2021-09-03

**Authors:** Soo-Eun Sung, Min-Soo Seo, Kyung-Ku Kang, Joo-Hee Choi, Sijoon Lee, Minkyoung Sung, Kilsoo Kim, Gun Woo Lee, Ju-Hyeon Lim, Seung Yun Yang, Sang-Gu Yim, Seul-Ki Kim, Sangbum Park, Young-Sam Kwon, Sungho Yun

**Affiliations:** 1Laboratory Animal Center, Daegu-Gyeongbuk Medical Innovation Foundation (DGMIF), Daegu 41061, Korea; sesung@dgmif.re.kr (S.-E.S.); msseo@dgmif.re.kr (M.-S.S.); kangkk@dgmif.re.kr (K.-K.K.); cjh522@dgmif.re.kr (J.-H.C.); sjlee1013@dgmif.re.kr (S.L.); tjdalsrud27@naver.com (M.S.); kskim728@knu.ac.kr (K.K.); 2Department of Biomaterials Science (BK21 Four Program), Pusan National University, Miryang 50463, Korea; syang@pusan.ac.kr (S.Y.Y.); sg.yim0425@gmail.com (S.-G.Y.); 3College of Veterinary Medicine, Kyungpook National University, 80 Daehakro, Daegu 41566, Korea; 4Department of Orthopedic Surgery, Yeungnam University Medical Center, Yeungnam University College of Medicine, 170 Hyochung-ro, Daegu 42415, Korea; gwlee1871@gmail.com (G.W.L.); globaljh2019@gmail.com (J.-H.L.); 5New Drug Development Center, Osong Medical Innovation Foundation, Chungbuk 28160, Korea; 6Efficacy Evaluation Team, Food Science R&D Center, KolmarBNH CO., LTD., 61Heolleungro 8-gil, Seoul 06800, Korea; lovesshot@kolmarbnh.co.kr; 7Institute for Quantitative Health Science & Engineering (IQ), Michigan State University, Auditorium Road 775 Woodlot Drive, East Lansing, MI 48824, USA; spark@msu.edu; 8Division of Dermatology, Department of Medicine, College of Human Medicine, Michigan State University, Auditorium Road 775 Woodlot Drive, East Lansing, MI 48824, USA; 9Department of Pharmacology and Toxicology, College of Human Medicine, Michigan State University, 775 Woodlot Drive, East Lansing, MI 48824, USA; 10Department of Veterinary Surgery, College of Veterinary Medicine, Kyungpook National University, Daegu 41566, Korea

**Keywords:** feline, mesenchymal stem cells, exosomes, extracellular vesicles, anti-inflammation

## Abstract

Adipose tissue-derived mesenchymal stem cells (AD-MSCs) release extracellular vesicles such as exosomes, apoptotic bodies, and microparticles. In particular, exosomes are formed inside cells via multivesicular bodies (MVBs), thus their protein, DNA, and RNA content are similar to those of the parent cells. Exosome research is rapidly expanding, with an increase in the number of related publications observed in recent years; therefore, the function and application of MSC-derived exosomes could emerge as cell-free therapeutics. Exosomes have been isolated from feline AD-MSCs and feline fibroblast cell culture media using ultracentrifugation. Feline exosomes have been characterized by FACS, nanoparticle tracking analysis, and transmission electron microscopy imaging. Moreover, cytokine levels were detected by sandwich enzyme-linked immunosorbent assay in exosomes and LPS-induced THP-1 macrophages. The size of the isolated exosomes was that of a typical exosome, i.e., approximately 150 nm, and they expressed tetraspanins CD9 and CD81. The anti-inflammatory factor IL-10 was increased in feline AD-MSC-derived exosomes. However, pro-inflammatory factors such as IL-1β, IL-8, IL-2, RANTES, and IFN-gamma were significantly decreased in feline AD-MSC-derived exosomes. This was the first demonstration that feline AD-MSC-derived exosomes enhance the inflammatory suppressive effects and have potential for the treatment of immune diseases or as an inflammation-inhibition therapy.

## 1. Introduction

Mesenchymal stem cells (MSCs) have multipotency as they can be differentiated to adipocytes, osteocytes, chondrocytes, etc. These cells are isolated from umbilical cord, adipose, and bone-marrow tissues [1]. In particular, adipose tissue can be easily and largely isolated from fat compared with other-site tissues [2]. Therefore, many studies have reported the cellular therapeutic effects of adipose tissue-derived mesenchymal stem cells. MSCs are well known for their characteristics and differentiation potential and exhibit many tissue-regeneration effects [3]. Therefore, mesenchymal stem cells with multiple lineages have been applied for damage regeneration in various tissues, such as the skin, cartilage, bone, adipose tissue, and muscle [4]. MSCs are very effective in reducing inflammation and play antioxidant roles in autoimmune disorders, such as that observed in a rheumatoid arthritis rat model [5]. A large number of experimental studies of tissue regenerative medicine have shown that MSCs secrete many factors that modulate angiogenesis for tissue regeneration or anti-inflammation [4]. Conversely, cytokines in AD-MSCs improved the symptoms of liver fibrosis. Because AD-MSCs express the angiogenic factors such as epidermal growth factor, vascular endothelial growth factor, and hepatocyte growth factor, they promote the proliferation of hepatocytes. Therefore, it seems that the therapeutic effects of mesenchymal stem cells are exerted via secreted growth factors, anti-inflammatory effects, and angiogenic effects [6]. Despite the observation that MSCs have various pleiotropic effects on tissue repair and great therapeutic effects on many diseases, cell free-based therapies have appeared as promising treatment strategies for various diseases [7]. Recent studies have shown that small extracellular vesicles derived from mesenchymal stem cells promote the anti-inflammation response and play a crucial role in reducing the inflammation response [8]. These extracellular vesicles (EVs) include exosomes, microparticles, membrane particles, and apoptotic vesicles. They are released from various cells under normal pathological conditions; therefore, they contain genetic information [9]. Microparticles and apoptotic vesicles are generated by shedding from the plasma membrane or budding from cells. The microparticles are 500–1000 nm in diameter, whereas apoptotic vesicles have a broad diameter of ~5000 nm; in contrast, exosomes are generated by intracellular vesicular trafficking and secreted through multivesicular bodies (MVBs) and also have a cup-shaped morphology and a heterogeneous size range (40–150 nm) [10,11]. Exosomes contain various mRNAs, miRNAs, and cytokines, and EVs are involved in cell-to-cell communication for tissue regeneration. Moreover, the inhibition of specific miRNAs improved their therapeutic response, thus they may act as a cancer therapy. Therefore, the factors included in exosomes can provide information regarding and have effects on cancer [12]. Recently, exosomes were used as drug-delivery systems for neurodegenerative diseases or cancer therapy across biological membranes [13,14]. Exosome isolation from many biofluids, such as serum, plasma, cerebrospinal fluid (CSF), urine, and saliva is possible, rather than exclusively from cell culture supernatants [15,16,17]. In this work, we isolated exosomes from cell culture medium to compare feline AD-MSCs and dermal fibroblasts. The exosomes were analyzed for the type of cytokine and growth factor expression levels using enzyme-linked immunosorbent assay. Moreover, we confirmed the differences in inflammation-related cytokines between feline AD-MSCs and feline dermal fibroblasts. Finally, these exosomes were tested on LPS-stimulated THP-1 macrophages for inflammatory response. Lipopolysaccharides (LPS) trigger a systemic inflammatory response by upregulating pro- and anti-inflammatory cytokines in THP-1 macrophages [18]. The production levels of the TNF-α, IL-1β, and IL-10 cytokines were different between the macrophages treated with feline fibroblasts and feline AD-MSC-derived exosomes. This study confirmed that feline AD-MSC-derived exosomes have potential in feline inflammatory diseases by reducing the inflammatory response.

## 2. Materials and Methods

### 2.1. Animals

The feline adipose tissue and skin tissue were obtained using female cats (*n* = 2, 4~5 years, mixed breed) employed as donors, who visited Kyungpook National University Veterinary Medical Teaching Hospital. Adipose tissues from inguinal fat pad and skin tissue from same animal under sterile and anesthesia conditions. The cats were pre-medicated with 0.1 mg/kg acepromazine maleate (Samwoo Medical, Yesan, Korea), and then 5 mg/kg propofol (Myeongmun Pharmaceutical, Seoul, Korea) was injected intravenously to induce anesthesia. Isoflurane (Hana Pharmaceutical, Hwasung, Korea) was used to maintain anesthesia. Biochemical and hematological analysis was conducted on the cats to confirm healthy condition prior to the sample collection.

### 2.2. Cell Isolation and Culture

Feline fibroblasts and feline AD-MSCs were derived from feline skin tissue and abdominal adipose tissue, respectively. In brief, the skin and adipose tissue were separated and rinsed in ice-cold phosphate buffered saline (PBS), followed by digestion in 0.45 µm filtered collagenase type I (2 mg/mL; Gibco, Invitrogen, Carlsbad CA, USA, 17018029) at 37 °C for 30 min. The solution was added to a 70 µm strainer and filtered, followed by centrifugation at 3000× *g* for 5 min. Feline fibroblasts and feline AD-MSCs were maintained in low-glucose Dulbecco’s Modified Eagle Medium (Thermo Fisher Scientific, Gibco, Carlsbad, CA, USA) supplemented with 10% fetal bovine serum (FBS) (Thermo Fisher Scientific, Gibco, Carlsbad, CA, USA) and 1% penicillin/streptomycin (Thermo Fisher Scientific, Gibco, Carlsbad, CA, USA), 10 µg/mL of recombinant human FGF-basic (Peprotech, Seoul, South Korea), 10 µg/mL recombinant human PDGF-BB (Peprotech, Seoul, South Korea), and 25 µg/mL plasmocin (InvivoGen, San Diego, CA, USA). All cells were cultured at 37 °C in 5% CO_2_. THP-1 cells (TIB-202;ATCC, TIB-202) were maintained in the base medium RPMI-1640 (ATCC), and the following components were added: 0.05 mM 2-mercaptoethanol (Life Technologies, Carlsbad, CA, USA), 10% FBS (Thermo Fisher Scientific, Gibco, Carlsbad, CA, USA), and penicillin/streptomycin to a final concentration of 1%.

### 2.3. Isolation of Feline Exosomes

The culture media from feline cells grown in 175 T-flasks were collected after 48 h of culture. Exosome-depleted FBS (Gibco) was used to prevent the isolation exosomes from other origins. Exosomes were purified from the cell culture medium and centrifuged for 10 min at 300× *g* to remove cells. The supernatant was centrifuged again for 25 min at 2500× *g* to remove cell debris and apoptotic bodies. Subsequently, the supernatant was ultracentrifuged for 120 min at 100,000× *g* at 4 °C using a type 90Ti rotor (Beckman Coulter, Brea, CA, USA) [16,19,20]. Exosome protein concentration was measured by a Pierce™ Bicinchoninic Acid assay kit (Thermo Fisher scientific, Rockford, IL, USA).

### 2.4. Flow Cytometry

Feline AD-MSCs were visualized using bright-field microscopy followed by flow cytometry of cell surface markers. Flow cytometry analyses were performed by the use of FACS (Gallios Flow Cytometer; Beckman Coulter, Brea, CA, USA). To determine the expression stem cell markers, cells were stained with antibodies such as anti-CD105 (Bio-Rad, Hercules, CA, USA, MCA1557), anti-CD90 (BioLegend, San Diego, CA, USA, 555596), anti-CD44 (103024; BioLegend, San Diego, CA, USA), anti-CD45 (555482; BioLegend, San Diego, CA, USA), anti-CD34 (343504; BioLegend, San Diego, CA, USA), and anti-CD14 (MCA1568; Bio-Rad, Hercules, CA, USA, MCA1568). The antibodies were conjugated with fluorescein isothiocyanate (FITC) or phycoerythrin fluorescent dyes. To determine feline AD-MSC-derived exosomes and feline fibroblasts-derived exosomes for flow cytometry, we used bead-coupled exosomes [16,21,22,23,24]. Isolated exosomes (100 µL) were incubated with 10 µL of aldehyde/sulfate latex beads for 15 min at room temperature, followed by the addition of 1 mL of PBS (supplemented with 0.1% BSA) into the exosome/bead mixture. Samples were incubated overnight with rotation. Bead-coupled exosomes were pelleted by centrifugation at 2000× *g* for 10 min and washed with 500 µL of PBS. The pellet was re-suspended in 50 µL of PBS containing anti-CD9 (NBP1-28364; NOVUS, Centennial, CO, USA) and anti-CD81 (NBP1-44859; NOVUS, Centennial, CO, USA, NBP1-44859) antibodies for 1 h at 4 °C. All antibodies were conjugated with FITC. Samples were washed using 500 µL of PBS and centrifuged at 2000× *g* for 10 min. The pellet was re-suspended in 150 µL of PBS. Gating of exosome-decorated beads with a diameter of 4 µm was performed and the results were analyzed. Data analyses were performed using Kaluza software version 2.1.1 (Beckman Coulter, Brea, CA, USA).

### 2.5. Transmission Electron Microscopy (TEM)

Freshly isolated exosomes from feline AD-MSCs and feline fibroblasts were re-suspended in cold distilled water. Exosome suspensions were loaded onto a formvar carbon-coated grid (Ted Pella Inc., Redding, CA, USA) and fixed in 2% paraformaldehyde for 10 min, followed by the removal of the solution and drying of the sample. Grids were observed using a bioTEM instrument (HT7700; Hitachi, Tokyo, Japan).

### 2.6. Nanoparticle Tracking Analysis (NTA)

Exosomes were diluted in PBS to a final volume of 1 mL. An NTA using a PMX120 (Particle Metrix, Ammersee, Germany) nanosight instrument was performed according to the manufacturer’s instructions.

### 2.7. Effects of Exosomes on LPS-Stimulated THP-1 Macrophages

Human monocytic leukemia THP-1 (50,000) cells were cultured in 12-well plates with 100 nM PMA (P1585; Sigma-Aldrich, St. Louis, MO, USA) for 24 h. Differentiated THP-1 macrophages were stimulated with 1 µg/mL LPS and simultaneously treated with 100 µg of feline fibroblasts-derived exosomes and feline AD-MSC-derived exosomes.

### 2.8. Cytokine Assays

Cytokines were quantified in exosome solution and THP-1 cell culture media by cytokine array (RayBiotech, Peachtree Corners, GA, USA). Subsequently, exosomes were diluted to approximately 250 µg/mL for each array, 100 µL of total sample were loaded, and analysis was performed according to the manufacturer’s protocol. The signals were measured by a laser scanner equipped with a Cy3 Wavelength Innoscan (Innopsys, Carbonne, France). Data were quantified using Mapix version 7.2.0.

### 2.9. Statistical Analysis

The statistical analyses were performed using the GraphPad Prism software. All measurement data were presented as means ± standard deviations and were obtained using an unpaired *t*-test. *p* < 0.05 was considered significant and is indicated in the figure legends.

## 3. Results

### 3.1. Characterization of Feline AD-MSC

The isolated feline AD-MSCs possessed typical MSC characteristics. Feline AD-MSCs adhered to plastic culture dishes and exhibited a spindle shape similar to fibroblasts (Figure 1A). In addition, feline fibroblasts were isolated from cat skin for control of stem cells [25,26,27,28]. Cells were maintained over 10 passages. Feline AD-MSCs were positive for the cell-specific surface markers CD105, CD90, and CD44, but were negative for CD14, CD34, and CD45, as assessed by flow cytometry analysis (Figure 1B). These results revealed that the isolated feline AD-MSCs had the classical characteristics of MSCs [29,30].

### 3.2. Isolation of Feline Fibroblasts and Feline AD-MSC-Derived Exosomes

Exosomes were isolated from feline fibroblasts and feline AD-MSC culture supernatant by ultracentrifugation. To confirm the purification of the exosomes, transmission electron microscopy (TEM), FACS, and nanoparticle tracking analysis (NTA) analyses were performed. According to TEM imaging results, the isolated exosomes exhibited the classical morphology of exosomes. They had a spherical shape with a diameter of 100–200 nm. Moreover, a dark and thick membrane was detected in the TEM images, implying that they had a lipid bilayer membrane [31] (Figure 2A). Exosomes prepared through MVB express several biomarkers such as tetraspanins, fusion proteins, and MVB biogenesis markers [32,33]. Tetraspanins are expressed outside of the exosome membrane, thus we detected tetraspanins, such as CD9 and CD81, using flow cytometry. Flow cytometry analyses are generally performed to assess cells; however, because exosomes are much smaller than cells, we used aldehyde/latex beads with a diameter of ~4 µm to detect exosomes in the flow cytometry analyzer [16,21,22]. Feline fibroblasts-derived exosomes showed expression levels of CD9 (11.7%) and CD81 (11.8%). In addition, feline AD-MSC-derived exosomes showed expression levels of CD9 (78.1%) and CD81 (28.5%) (Figure 2B). The feline fibroblast-derived exosomes showed lower expression of CD9 and CD81 than the feline AD-MSC-derived exosomes. Thus, the expression of tetraspanins on the surface of exosomes differed depending on the cell type [23,34,35]. The NTA data revealed the exosome size distribution and concentration in the isolated exosomes. The average diameter of feline fibroblast-derived exosomes was 156.4 nm and their concentration was 2.3 × 10^10^ particles/mL (Figure 2C), whereas the average diameter of feline AD-MSC-derived exosomes was 155.4 nm and their concentration was 1.2 × 10^10^ particles/mL (Figure 2C).

### 3.3. Feline AD-MSC-Derived Exosomes Have Immunosuppressive Functions

The immunosuppressive properties of mesenchymal stem cells have been reported; therefore, we compared the cytokine and chemokine levels between feline AD-MSC-derived exosomes and feline fibroblast-derived exosomes. To compare the measured levels of inflammatory factors exactly, we adjusted protein concentrations to 250 µg/mL, as recommended by the cytokine array manual. The detection of several cytokines and chemokines indicated the presence of pro-inflammatory factors, such as IL-1 β, IL-2, IFN-gamma, IL-8, and the chemokine RANTES. These pro-inflammatory factors were expressed at low levels in feline AD-MSC-derived exosomes compared with feline fibroblast-derived exosomes. However, the anti-inflammatory factor IL-10 was significantly upregulated in feline AD-MSC-derived exosomes compared with feline fibroblast-derived exosomes. Therefore, MSC-derived exosomes played a crucial role in the immune defense compared with fibroblast-derived exosomes (Figure 3).

### 3.4. Anti-Inflammatory Effects of Feline AD-MSC-Derived Exosomes on THP-1 Macrophage Cells

Macrophages secrete many cytokines and chemokines when stimulated by antigens such as lipopolysaccharide (LPS) or concanavalin A (ConA) [18]. The human leukemia monocytic cell line THP-1 was differentiated to macrophage-like cells by treatment with Phorbol 12-myristate 13-acetate (PMA) (Figure 4A). The floating cell THP-1 was attached to the cell culture dish due to PMA treatment. The differentiated THP-1 macrophages were used to analyze the anti-inflammatory response elicited by exosomes. Feline AD-MSC- and fibroblast-derived exosomes were treated with 1 µg/mL of LPS (Figure 4A). THP-1 was damaged when LPS was treated due to the inflammatory response induced by LPS (Figure 4A). LPS-stimulated THP-1 cells expressed high levels of cytokines, such as TNF-α, IL-1β, and IL-10 (Figure 4B). Moreover, TNF-α was decreased significantly after treatment with LPS and feline AD-MSC-derived exosomes simultaneously (186.75 ± 26.45 pg/mL) compared with treatment with LPS alone (336.80 ± 119.55 pg/mL) or LPS together with feline fibroblast-derived exosomes (373.04 ± 59.25 pg/mL). Another pro-inflammatory factor, IL-1β, exhibited a pattern similar to that of TNF-α. IL-1β production was increased in the group treated with LPS alone (669.45 ± 124.82 pg/mL). In turn, IL-1β production was increased in the group treated with LPS plus feline fibroblast-derived exosomes (774.00 ± 22.54 pg/mL) but was significantly blocked after the administration of LPS plus feline AD-MSC-derived exosomes (536.44 ± 121.92 pg/mL). In contrast, the anti-inflammatory factor IL-10 was not significantly changed in the group treated with feline AD-MSC-derived exosomes (230.01 ± 25.14 pg/mL) compared with the groups treated with LPS alone (183.40 ± 66.28 pg/mL) and with LPS together with feline fibroblast-derived exosomes (159.55 ± 36.83 pg/mL). Meanwhile, we tried to treat feline fibroblasts exosomes or feline AD-MSC-derived exosomes without LPS. As a result, the expression of cytokines was lower than the control group without LPS and exosomes. Thus, it could not be displayed the data on the graph. These results suggest that feline AD-MSC-derived exosomes exert their anti-inflammatory effects via the downregulation of the pro-inflammatory factors TNF-α and IL-1β.

## 4. Discussion

EVs are classified according to their size distribution. To isolate 30–200 nm-sized exosomes, we performed serial centrifugation. There are many different methods to isolate exosomes, however the method using centrifugation or ultracentrifugation is low in cost and a reasonable approach to the separation of exosomes; moreover, it has good harvest efficiency from mass cell culture media compared with other isolation methods [19,20,36]. Although the method using ultracentrifugation to isolate exosomes is used most widely, it is necessary to discuss its efficacy regarding yield and particle aggregation during the isolation of exosomes [37,38]. In our results, the isolated feline AD-MSC-derived exosomes reduced the effects of inflammation via the low expression of pro-inflammatory cytokines and chemokine secretion, and expression of high levels of the anti-inflammation factor IL-10. AD-MSC-derived exosomes have effects on the anti-inflammatory potential and obesity because they acquire the ability to shift macrophages from the M1 to the M2 type via macrophage polarization [39,40]. Moreover, recent studies have emphasized the role of exosomes in angiogenesis, anti-inflammation, and wound healing [41,42]. IL-1β is a pro-inflammatory cytokine that plays a critical role in pain, chronic inflammation, and autoimmune disorders. IL-1β is secreted from fibroblasts and endothelial, neuronal, and immune cells, such as macrophages and mast cells [43]. The pro-inflammatory cytokine IL-2 controls inflammatory diseases. IL-2 acts as an inducer of T-cell proliferation and as a Th1 and Th2 effector [44]. In turn, IL-8 activates neutrophils at inflammatory sites and acts as a causative cytokine in acute inflammation by recruiting neutrophils [45,46]. Interferon gamma (IFN-γ) induces an inflammatory response and apoptotic cell death; however, several reports have stated that IFN-γ deficiency activates inflammatory signals. Therefore, the regulation of IFN-γ may have potential in the treatment of rheumatoid arthritis [47]. IL-10 helps inhibit the Th1 effector for host immune response to pathogens; thus, dysregulation of IL-10 causes immunopathology and increases autoimmune diseases [48,49]. IL-10 plays a central role in infection and has anti-inflammatory properties. Therefore, the manner in which IL-10 exerts regulation in immune cells will be important as it may negatively regulate pro-inflammatory cytokines [50,51]. RANTES is a chemokine that enhances the inflammatory cascades [52]. The chemokine RANTES participates in the regulation of perivascular inflammation, vascular malfunction, and T-cell accumulation in cardiovascular disease [53]. Recently, exosomes and secretomes were separated from feline AD-MSCs, and proteomics KEGG and GO analyses were conducted to address the possibility of using exosomes as therapeutic substances [54]. Several studies reported the analysis and characterization of exosomes derived from canine-species MSCs and presented differential expression of cytokines and growth factors in adipose tissue- and bone-marrow-derived MSC exosomes [55]. Research studies using canine reported the comparison of miRNA expression levels in exosomes according to the sources of biofluids, such as serum and urine [56]. Stem cell-derived exosomes not only exert immune defenses, but also play a crucial antitumor role [57,58,59]. MSC-derived exosomes can be used as cell-free therapeutics because of their immunomodulatory and regenerative effects [60]. However, further studies would be necessary for its therapeutic effects with diverse diseases animal models to confirm the clinical efficacy, treatment route, and dose, as well as its safety. Though our data has a limitation by lack of in vivo studies for feline species, feline AD-MSC-derived exosomes can be a good candidate for cell free-based therapies for clinical application.

## 5. Conclusions

To separate purified exosomes from feline fibroblasts and feline AD-MSCs cell culture medium, we proceeded with centrifugation in stages. The first was to remove cell debris, the second was to remove microparticles, and finally to separate exosomes ultracentrifuged at 100,000× *g*. As a result, the cup-shape morphological 100–200 nm-size exosomes could be isolated. Exosomes contain various DNAs, RNAs, and proteins that are involved in cell-to-cell communication for immune response, tissue regeneration, and therapeutic effects. We identified the amount of cytokine, an immune substance contained in exosomes isolated from feline cells. As a result, several cytokines were able to identify significant differences between fibroblasts-derived exosomes and AD-MSC-derived exosomes among them, known as pro-inflammatory factors and anti-inflammatory factors. However, the amount of cytokine in the exosomes themselves was very small, so we tested in immune cells for anti-inflammatory effects. Anti-inflammatory effects were compared when LPS was treated in macrophages to induce an inflammatory response and exosomes were treated in these cells. As a result, it was confirmed that exosomes had an anti-inflammatory effect by decreasing TNF-α and IL-1β and increasing IL-10 when treated. In particular, it was confirmed that AD-MSC-derived exosomes are more effective than fibroblasts-derived exosomes. These results show the possibility of using feline AD-MSC-derived exosomes as a treatment for inflammatory diseases in pets such as cats.

## Figures and Tables

**Figure 1 vetsci-08-00182-f001:**
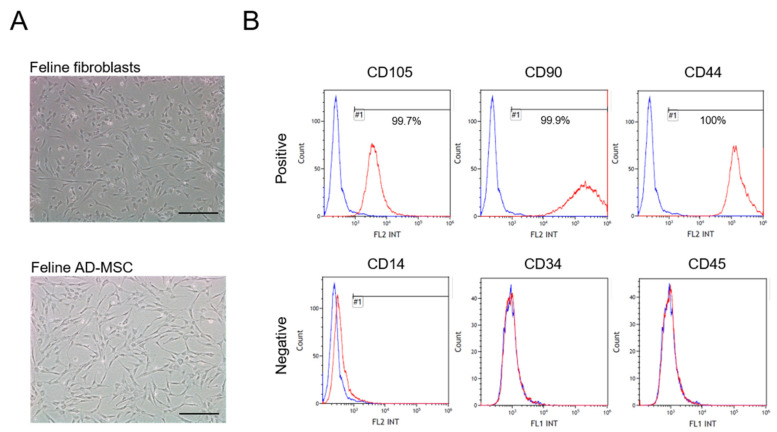
Characterization of feline AD-MSCs. (**A**) The morphology of the isolated feline fibroblasts (upper) and feline AD-MSCs (lower) was assessed microscopically (scale bar = 25 µm). (**B**) Feline AD-MSCs were analyzed positive (CD105, CD90, and CD44) and negative (CD14, CD34, and CD45) stem cell markers by flow cytometry. Blue line of the histograms indicated unstained control. Red line indicated stained cells. (X-axis: FL1 is FITC intensity, FL2 is PE intensity).

**Figure 2 vetsci-08-00182-f002:**
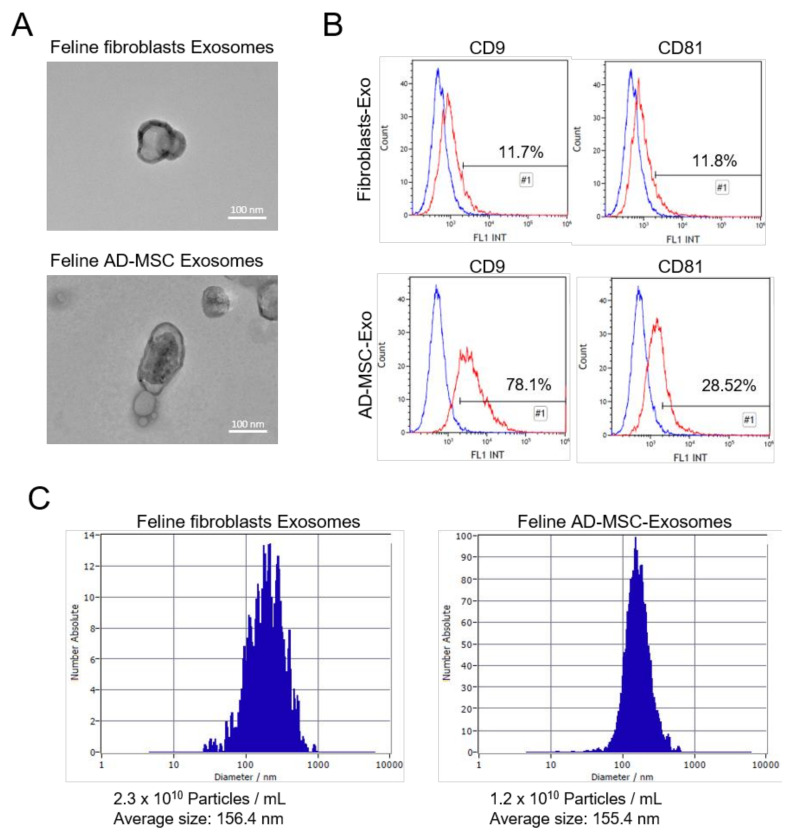
Identification of feline fibroblast- and feline AD-MSC-derived exosomes. (**A**) Electron micrographs of isolated exosomes showing exosomes with a typical morphology and size. (**B**) The exosome surface markers tetraspanins (CD9 and CD81) were detected by flow cytometry. (**C**) Nanoparticle tracking analysis (NTA) was used to assess the concentration, average size, and size distribution of the isolated exosomes.

**Figure 3 vetsci-08-00182-f003:**
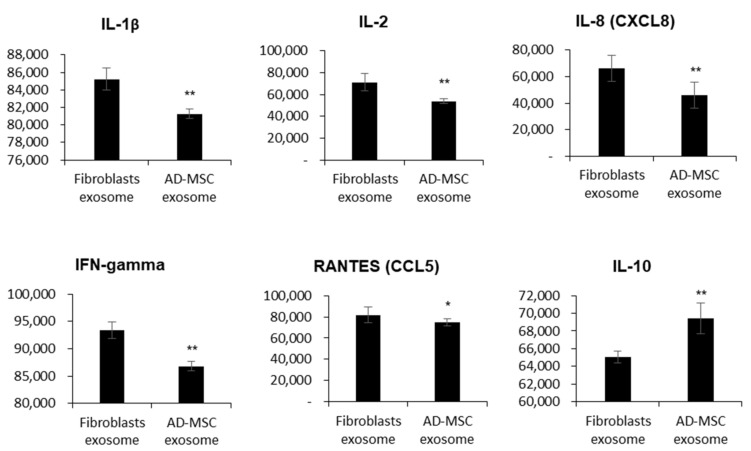
Comparison of cytokine and chemokine levels between feline fibroblast-derived exosomes and feline AD-MSC-derived exosomes. The Y-axis indicates fluorescence intensity (a.u), * *p* < 0.05, ** *p* ≤ 0.0001.

**Figure 4 vetsci-08-00182-f004:**
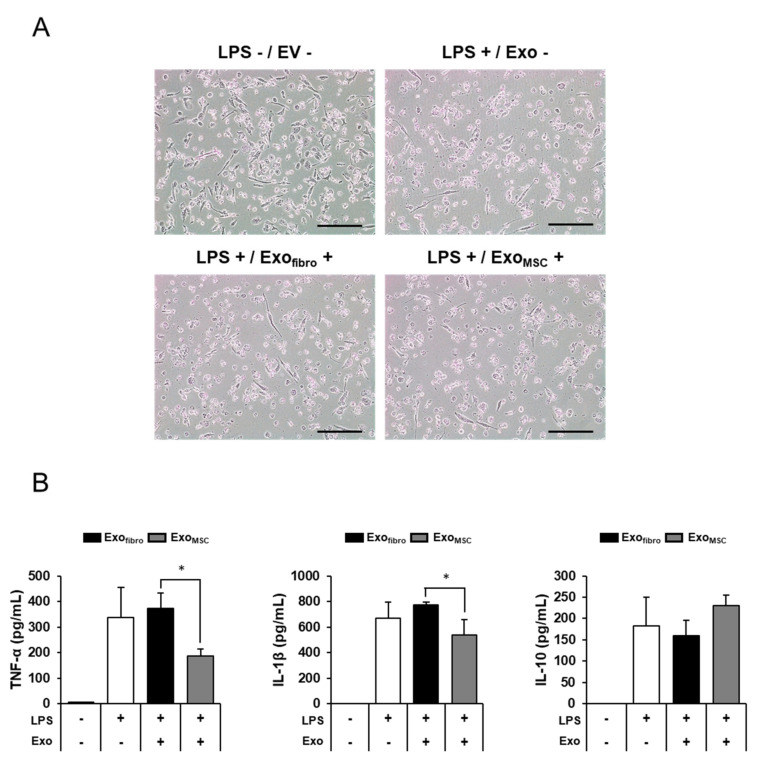
Anti-inflammatory effects of feline fibroblast- and feline AD-MSC-derived exosomes on human macrophages. (**A**) Human monocytic leukemia THP-1 cells were differentiated to macrophages using 100 nM PMA. THP-1 cells were stimulated with LPS and treated with 100 µg of feline cell-derived exosomes (scale bar = 25 µm). (**B**) The pro-inflammatory factors TNF-α and IL-1β and the anti-inflammatory factor IL-10 were measured after treatment with LPS and/or exosomes (Exo) for 24 h (* *p*-value < 0.05).

## Data Availability

The data that support the findings of this study are available from the corresponding author upon reasonable request.

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
