# Peer review of "Mesenchymal Stem Cell Exosomes Derived from Feline Adipose Tissue Enhance the Effects of Anti-Inflammation Compared to Fibroblasts-Derived Exosomes"

_vetsci, 2021, doi:10.3390/vetsci8090182_

Round 1

Reviewer 1 Report

In this article, authors investigated the immunomodulatory effect of adipose tissue derived MSC’s exosomes in cats. Accumulation of this type of research remains important for the development of the new therapy for immune diseases. However, the author needs to clarify some important points before publication.

  1. 1, Only surface maker analysis for MSC characterization is not adequate. Authors need to provide more detailed analysis at least CFU ability, multiple differentiation ability.
  2. 2, Panel B, The expression levels of CD9 and CD81 in Fibroblast exosomes are looks not so high compared to isotopic control. Authors need to clarify the cut off value (#1 in the flow data, Usually, 5% or 1% are well accepted.).
  3. 4, Panel A, The difference of 4 picture is not so clear. Author need to provide the detailed description about these data including evaluation methods. Panel B, Authors need to provide the data about cytokine levels of exosomes only, otherwise, base line is totally not clear.

Reviewer 2 Report

The title should mention feline fibroblast, as the study is not only based on MSCs. 

The exosome from MSCs were characterized by flow cytometer but the one isolated from fibroblast no information is provided in methods. 

THP1 cells were treated with 100ug of feline fibroblast derived exosomes but not with MSCs derived exosomes. 

Figure 1 legend should mention feline fibroblast. 

References need to be updated. 

Reviewer 3 Report

The authors describe in their manuscript the effect of feline adipose mesenchymal stem cell derived exosomes towards an antiinflammatory property. In their results the authors indicate a similar behaviour of cell supernatants as already described for other animals or humans. The work is interesting as it shows a species independent function and would allow the use of therapeutics developed in humans also in cats.

However, two major concerns should be resolved:

Concern one:

The authors use latex beads with a size of 4um, an appropriate size for detecting vesicles should be used.

Concern two:

The authors measure cytokines as one of their main read outs. However, the authors need to demonstrate that the cytokines are indeed connected to the vesicle and not remnants of the isolation process.

Round 2

Reviewer 1 Report

Authors provided proper answers in revised manuscript. 

Reviewer 3 Report

Authors tried to answer questions